# Miniature Integrated 2.4 GHz Rectennas Using Novel Tunnel Diodes

**DOI:** 10.3390/s23146409

**Published:** 2023-07-14

**Authors:** Christopher Walsh, Saad G. Muttlak, Mohammadreza Sadeghi, Mohamed Missous

**Affiliations:** 1Department of Electrical and Electronic Engineering, University of Manchester, Manchester M13 9PL, UK; 2Advanced Hall Sensors Ltd., Manchester M17 1RW, UK; mohammadreza.sadeghi@ahsltd.com

**Keywords:** asymmetrical spacer layer tunnel (ASPAT) diode, biomedical implant devices, electrically small antennas, impedance matching, planar antennas, rectennas, wireless power transfer

## Abstract

This work presents the design, fabrication, and measured results of a fully integrated miniature rectenna using a novel tunnel diode known as the Asymmetrical Spacer Layer Tunnel (ASPAT). The term rectenna is an abbreviation for a rectifying antenna, a device with a rectifier and antenna coexisting as a single design. The ASPAT is the centrepiece of the rectifier used for its strong temperature independence, zero bias, and high dynamic range. The antenna is designed to be impedance matched with the rectifier, eliminating the need for a matching network and saving valuable real estate on the gallium arsenide (GaAs) substrate. The antenna is fully integrated with the rectifier on a single chip, thus enabling antenna miniaturisation due to the high dielectric constant of GaAs and spiral design. This miniaturisation enables the design to be fabricated economically on a GaAs substrate whilst being comparable in size to a 15-gauge needle, thus unlocking applications in medical implants. The design presented here has a total die size of 4 × 1.2 mm^2^, with a maximum measured output voltage of 0.97 V and a 20 dBm single-tone 2.35 GHz signal transmitted 5 cm away from the rectenna.

## 1. Introduction

The rectenna has seen widespread applications in various fields, consisting of a rectifier and an antenna. This sees applications where there is a need for wireless power transfer (WPT), energy harvesting (RF-EH), information transfer, or simultaneous wireless information and power transfer (SWIPT). The most notable application in the literature is William Brown’s “Microwave Powered Helicopter” [1], which is the first recorded use of a rectenna. More recently, the European Space Agency’s SOLARIS project announced plans to use a rectenna as part of a space-based solar power system [2], where solar panels will generate electricity 24 h a day and beam it to earth using microwaves to be converted to electricity using the rectenna.

There are also further applications of rectennas to harvest energy in the optical and infrared spectrum, where, given the higher frequency, nanoscale rectennas are required. One example uses carbon nanotubes (CNT) as a dipole antenna with single, double [3], or quad layers of insulators to fabricate a CNT–insulator–metal rectifier with a reported conversion efficiency of 3 × 10^−6^% in the case of a quad-insulator diode [4]. Further examples use a metal–insulator–metal rectifier with a 28.3 THz bow-tie antenna to achieve an efficiency of 3.7 × 10^−8^% [5], or a 280 THz nano-cone antenna with an efficiency of 2.3 × 10^−4^% [6].

However, the main objective of this work is to miniaturise RF rectennas to a millimetre scale, thus unlocking their potential for use in medical implants. In healthcare settings, needles with sizes from 30- to 18-gauge (0.159 to 0.838 mm inner diameter) are readily available [7,8], while larger 11-gauge needles (2.388 mm inner diameter) are used for bone marrow biopsies [9]. As such, miniaturising rectennas to this scale would allow them to be implanted using standard medical equipment, whereas miniaturisation to the smallest needle size possible would be preferred to reduce discomfort and bleeding [10].

The rectenna plays a role in intelligent medical implants as a power source or communication link. Specific applications of these implants are presented in [11], where some notable examples are glaucoma sensors, intracranial pressure monitoring, and retinal implants.

Antenna miniaturisation presents limited bandwidth and efficiency when the antennas are electrically small. These electrically small limitations are defined by the Chu–McLean limit when the condition ka<1 is true. In this condition, k=2π/λ and *a* is the radius of the antenna. To provide an example of this limitation, any 2.4 GHz antenna with a radius under 20 mm would be considered electrically small, and the maximum attainable gain for a lossless antenna is defined by
(1)$$G={\left(ka\right)}^{2}+2\left(ka\right).$$

Using (Equation 1) [12], it can be calculated that the maximum attainable gain for a 2 mm radius 2.4 GHz antenna is −6.75 dBi. Due to ”2(ka)” being the dominant component of (Equation 1), it is evident that the radius and the gain have a linear relationship at the mm scale. It would be advantageous to have a larger design, but to fabricate the on-chip antenna economically, it would not be feasible to have a design that has a radius of 2 cm, nor would it be practical for biomedical implant applications. In [13], a range of folded dipole antennas was designed from 1 × 3 to 1 × 10 mm^2^ in size, which demonstrated the Chu–McLean limit, in which the largest antenna was the most efficient design.

Despite these limitations, several systems in the literature have been proposed using a range of technologies to achieve the on-chip integration of rectifiers and antennas, and the most common being complementary metal-oxide semiconductor (CMOS) processes with a wide array of examples in [14,15,16,17,18,19,20,21], which have target applications in biomedical implants, radio frequency identification (RFID), or the Internet of Things. A range of different rectifier design choices is used in these works, such as a cross-connected full-wave CMOS rectifier, a general voltage double using a diode-connected CMOS transistor, or a Dickson rectifier. Compound semiconductor technologies have also been attempted, where in [22], a gallium arsenide (GaAs) pseudomorphic high-electron-mobility transistor (pHEMT) was used as a diode in a voltage doubler circuit, and [13] used the voltage double circuit with an Asymmetrical Spacer Layer Tunnel (ASPAT) diode, which is also used in the work presented here.

The miniaturisation of the antenna can be achieved by folding the dipole around the circumference of the die as a coil or folding each side of the dipole into a spiral. These methods maintain the same electrical length of the antenna while significantly reducing the die size of the whole circuit.

The circuit requires a rectifier to convert the RF signal into a DC voltage to complete the rectenna design. In this work, the ASPAT diode, first proposed for use in microwave circuits in [23], performs this rectification due to its advantages over other diodes. These advantages are its inherent zero-bias property, which eliminates the need for bias circuitry, and its temperature independence. These advantages are due to the main transport mechanism being quantum tunnelling. This is due to the structure of the diode having a thin (2.83 nm) aluminium arsenide (AlAs) barrier sandwiched by two spaced GaAs, each with an unequal thickness at a ratio of 1:40. This asymmetry in the design of the diode leads to its name and its asymmetric IV characteristics.

The cross-section of this diode can be seen in Figure 1 with each region colour coded. Besides the aforementioned active region consisting of the barrier (red) and the spacers (brown), there are the lightly doped collector and emitter layers (light green) with heavily doped ohmic layers (dark green) for the anode and cathode contacts. The planar structure of this diode design makes it highly suited for integration in a rectifier circuit and a rectenna. The wafer of the ASPAT diode is grown through molecular beam epitaxy (MBE), after which the wafer is processed using standard i-line photolithography used to etch and deposit metal on the substrate, as detailed in [24,25].

In this work, the ASPAT diode was part of a rectifier circuit to convert the RF signal into a DC voltage. The diode was made available for this project through epitaxial wafers from prior projects [24], for which this work designed an monolithic microwave integrated circuit (MMIC) layout as part of the whole design to fabricate the structure for an ASPAT diode.

We can see the aforementioned advantages of the ASPAT diode summarised in Table 1 alongside other key diode parameters. This table compares the ASPAT diode against a silicon Schottky diode, a planar-doped barrier diode, and a backward (Esaki) diode using data from [26]. This table lists the temperature independence as the variation in output voltage from −40 to 80 °C, with the ASPAT diode only varying by 30% as opposed to the 120% seen in a Schottky diode. Alongside this key advantage, the ASPAT diode has high voltage sensitivity ($\beta_{v}$) and a high dynamic range, proving its suitability for various applications.

In this paper, a fully integrated miniature rectenna is designed using a co-design process to eliminate the need for a matching network. With the inclusion of a resistive stub, this matching can be achieved with a smaller die length. Including this resistive stub and optimising the resistive load shows that this design has the highest figure of merit when compared to prior works in the 2.4 GHz band.

## 2. Materials and Methods

The rectenna was designed using a combination of CST Studio Suite [27] and Keysight ADS [28], where CST was used to design the antenna and ADS was used to model the ASPAT, design the rectifier, and design the full layout of the rectenna.

As previously discussed, the rectenna is a combination of two elements, the rectifier and the antenna. One objective of the design was to eliminate the need for a matching network [29] to reduce the rectifier circuit’s area by removing the traditional 50 Ω boundary. Consequently, the antenna and circuit needed to be co-designed to ensure that their complex impedances had a conjugate match, thereby maximising power transfer. As a rectifier is typically highly capacitive, its impedance is measured as R−jX; as such, the antenna should be designed so that its impedance is R+jX. These techniques were applied to designs in [30,31,32].

This section will describe the design process of the rectifier, the antenna, and the whole design of the die. Due to the co-design process, several iterations of the rectifier and antenna were designed to optimise the die size and the impedance matching of the design.

### 2.1. Rectifier Design

The rectifier circuit in this design was used to convert the received RF signal into a DC output, using the rectifier’s nonlinear IV characteristics to detect the microwave signal [23]. The rectifier circuit can be designed using transistors or diodes; however, for this particular design, the focus is on diode-based circuits due to the utilisation of the ASPAT diode and its previously discussed advantages.

The rectifier circuit can be designed in several different ways, as discussed in [33]. A single diode could be used in a half-wave rectifier in either a shunt or series configuration. However, for this design, a general voltage doubler circuit is used, as per Figure 2. The general voltage doubler, also known as the Greinacher doubler, consists of two diodes that work together to multiply the output, theoretically producing double the output voltage of a half-wave rectifier. This design was also chosen due as it would allow for a simple transition from schematic to layout. It utilises two ASPAT diodes and one capacitor connected in series. Additionally, a capacitor and resistor were included in this design as the load of the circuit, forming an RC filter.

The ASPAT diode shown in Figure 2 uses an equivalent circuit model derived from the measured results, with the methodology discussed in [34,35]. The measured results are used to derive the parasitic and intrinsic equivalent circuit parameters. Using the equivalent circuit model, the derived impedance of the diode is given by
(2)$$Z_{11}=R_{s}+\frac{R_{j}}{1+{\left(\omega C_{j}R_{j}\right)}^{2}}-j\frac{\omega C_{j}R_{j}^{2}}{1+{\left(\omega C_{j}R_{j}\right)}^{2}}$$
where the series resistance ($R_{s}$), junction resistance ($R_{j}$), and junction capacitance ($C_{j}$) determine the impedance of the device. The model was then expanded to incorporate its measured DC-IV characteristics as discussed in [36,37].

The derived results for a 6 $\times \;6\;\mu m^{2}$ ASPAT diode are $R_{s}=22.68\;\Omega$, $R_{j}=39.20\;k\Omega$, and $C_{j}=61.82\;fF$, with the junction resistor replaced in the model by a ”symbolically defined device” in ADS. The ”symbolically defined device”, or ”SDD”, contains a polynomial equation that fits the measured I-V curve of the ASPAT diode. As a result, the model of the ASPAT consists of a SDD and 61.82 fF capacitor in parallel, with a 22.68 Ω resistor in series. The model is evaluated in Figure 3, where both the S-parameter and DC simulations show excellent agreement, showing that the model provides a good basis for the rectifier design.

To determine appropriate values for the series capacitor, load capacitor, and load resistor, the schematic of the circuit was simulated using a harmonic balance simulator in ADS. The parameter sweeps showed that the capacitors should be at least 2 pF in size and the resistor should be as large as possible to maximise the output voltage of the rectifier. However, simulations showed that the maximum power would be seen when the load was 7 kΩ. As a result of these choices, a compromise of 25 kΩ was chosen. This value was chosen as a design choice to balance power and voltage because 25 kΩ was located at the “knee” point of a voltage vs load resistance plot. The layout design assumed a titanium (Ti) resistive layer with 50 Ω/sq. This resistivity required 500 squares or a length of 2500 μm with a width of 5 μm.

The series and load capacitors were designed to be 2.9 and 7.5 pF, respectively, based on the assumption of a 240 nm silicon monoxide (SiO) dielectric film with a dielectric constant of 5.3. These sizes were chosen based on the size constraints of the layout, such that there was enough area for the bond pads, resistors, diodes, and interconnects, with the remaining area filled by these capacitors.

The schematic in Figure 2 was then simulated in ADS using the large signal S-parameter (LSSP) simulator, chosen due to the circuit’s impedance being dependent on the input power. The results of the LSSP simulation are shown in Figure 4a,b, where it is shown that the imaginary component of the impedance of the rectifier has only minor variation with its power level. However, the real component sees strong variation with the input power. As the impedance of the rectifier depends on the power delivered, the target power level was estimated to be 0 dBm, at which point the rectifier’s impedance is (53.5-j391.5) Ω. The variance in the imaginary component in Figure 4b at 2.4 GHz is from −392 to −379 Ω, whereas the real component in Figure 4a varies from 14.8 to 77.4 Ω across the power range considered.

### 2.2. Antenna Design

The options available for the antenna design were limited to dipoles due to the fabrication process, which only deposits metal on one side of the die. Consequently, a uniplanar design was chosen, where the signal and ground planes of the antenna are located on the same plane.

Typical dipole antenna designs are λ/2 in width, which at 2.4 GHz would equate to an antenna diameter of 6.25 cm. This size would be unfeasible for integration on a GaAs wafer. Further, the standard mask size for the fabrication procedure has a usable area of 12 × 12 mm^2^. To economically integrate a 2.4 GHz antenna on a GaAs wafer and realise a fully integrated circuit, antenna miniaturisation techniques must be used.

One key aim of the antenna design process is to achieve an impedance match without the use of a matching network to conserve space on the die. This goal can be evaluated using
(3)$$S_{11}=20\log_{10}\left(\frac{Z_{L}-Z_{S}^{*}}{Z_{L}+Z_{S}}\right)$$
where $S_{11}$ is the reflection coefficient in dB, $Z_{L}$ is the rectifier’s impedance, and $Z_{S}$ is the antenna impedance [38]. Equation (Equation 3) shows that for impedance matching to be achieved, the impedance of the antenna should be a conjugate to the rectifier, with the antenna structure chosen to help achieve this.

The main resonating arms of the antenna are folded into a spiral to minimise the physical size of the antenna with an outer resistive stub to improve the matching characteristics of the design, inspired by the designs presented in [39,40].

The final antenna design uses 12 segments in the spiral, with an additional trace around the outer perimeter. The number of spirals was chosen by incrementing the number of segments and optimising the trace width and gap until the minimum reflection coefficient was obtained. In total, this design had 6 parameters that had to be determined: the die width, the die length, the number of segments, the inner trace width (tw), the outer trace width (otw), and the trace gap (tg), all of which are illustrated in Figure 5.

The first parameter to be determined was the die length, with the aim to minimise the size of the rectenna whilst having acceptable antenna performance. The die length was chosen to be 4 mm due to it allowing full use of the 12 × 12 mm^2^ mask size. Simulations were performed with this die length to determine that the die width should be 1.2 mm, as this allowed the design to achieve good impedance matching whilst minimising die size.

### 2.3. Rectenna Design

The antenna design in CST was recreated in ADS using Application Extension Language (AEL) layout macros. This approach results in a parameterised layout, enabling further optimisation with the rectifier layout already implemented. The layout was then simulated and re-optimised using the momentum microwave simulator and the port had a complex impedance of (53.5-j391.5) Ω, with the optimisation aim to minimise $S_{11}$. To avoid optimising to an unnecessary precision, the simulation variables were set with a step size of 0.1 μm. The momentum simulation was set so that the mesh frequency was 2.4 GHz, the mesh density was 100 cells/wavelength, and the edge mesh had an auto-determined edge width. The substrate in momentum had a 650 μm GaAs (k = 12.9) substrate, followed by a 240 nm SiO (k = 5.3) dielectric layer and an 800 nm gold layer.

The parameterised layout was optimised for a range of die widths, and the chosen optimised result had a die width of 1.2 mm with 12 segments. The result of the momentum optimisation used the following optimised parameters: outer trace width (otw) = 75.6 μm, trace width (tw) = 48.4 μm, and trace gap (tg) = 88.1 μm.

This optimisation was then simulated using the LSSP (large signal S-parameter) to allow for the impedance matching across a range of power levels to be determined, and it is presented in Figure 6. In this figure, the simulated Z-parameters of the antenna and rectifier are combined as a calculated S-parameter, using (Equation 3), such that the S-parameters presented are not referenced to 50 Ω but to the impedance of the antenna. This simulation shows that the resulting antenna and rectifier are well matched at 2.4 GHz, particularly at higher power levels. For example, at 10 dBm, the simulated $S_{11}$ is −23.5 dB, and at 0 dBm, it is −11.7 dB. As a result, this design can be considered to be well matched at these power levels. However, when looking at the result at −10 dBm, the impedance match reduces to −5.9 dB, which, in mobile applications, would still be considered good [41], but operating at higher power levels would be desired for this design.

The results in Figure 6 show that the design is centred at 2.4 GHz across the power range considered. When referring to the previously simulated impedance of the rectifier, the imaginary component has little influence on the impedance matching of the design. Considering the imaginary component independently from −389 to −379 Ω with a real component of 53.5 Ω, the simulated S-parameter results at 2.4 GHz vary from −11.7 to −12.3; however, when considering only the real component from 14.8 to 77.4 Ω with a fixed imaginary component at −391.5 Ω, the variance is much stronger. The simulated results showed an impedance match of −2.9 to −18.8 at 2.4 GHz with the centre frequency remaining at 2.4 GHz, showing that this design is well tuned to the target frequency at the power ranges considered.

The new parameters were then used as part of the prior CST model and simulated to determine the far-field performance of the antenna. This is shown in Figure 7, where the maximum gain was found to be −24.7 dBi. While this is a low gain for an antenna, it is to be expected for a design that is approximately λ/30 in size.

### 2.4. Fabrication

The design was fabricated using standard i-line photolithography techniques and wet etching processes, of which the first 6 steps were used to define the structure of the ASPAT diode as detailed in [37]. Further, we used a 400 nm gold layer for the bottom plate of the capacitor, a 240 nm SiO dielectric, a 25 nm Ti resistive layer, and a 800 nm gold alloy (Ti/Au) for the top plate of the capacitor and the antenna. Finally, a passivation layer was deposited on top to protect the structures, with the measurement areas left exposed.

The mask contained several test structures to allow a greater understating of the fabrication process, and included diodes, resistors, and capacitors. The measurements of the diodes confirmed that the diodes functioned well and had comparable performance to prior fabrication runs, such that the models used in Section 2.1 would be valid for this fabrication run.

Further, the capacitor measurements showed that the capacitor dielectric had a dielectric constant of 5.3, and the resistors showed that the resistors had a sheet resistance of 96 Ω/sq. This difference between the assumed value of 50 Ω/sq and the measurement is due to the difficulty in controlling how much of the material is oxidised during fabrication. These results differed from the initial simulation but were expected to have little impact on the results of the full rectenna. Additionally, two antennas were fabricated without a connection to the rectifier. This was to allow for the on-wafer measurement of the antenna on its own.

A fully fabricated rectenna is shown in Figure 8, where a high-resolution optical micrograph was taken. An S-parameter measurement of the device was taken using GS (ground signal) on-wafer probes from the bottom contacts, and the large square contacts were bonded for DC measurements.

## 3. Results

The first set of measurements that took place on the rectennas was an S-parameter measurement using a calibrated vector network analyser (Anritsu 37369A VNA) through the short-open-load-thru (SOLT) method with a reference impedance of 50 Ω. The measurements were then translated into real and imaginary Z-parameter results, presented in Figure 9a,b respectively.

Figure 9a,b both indicate that the fabricated antennas resonate at a lower frequency and magnitude compared to the simulation. Consequently, the frequency at which a peak voltage occurs is lower than 2.4 GHz, resulting in a reduced impedance match when compared to the simulated results, as shown in Figure 6.

Two of the fabricated rectennas were then bonded to a SOP8 IC adapter [42] with wires soldered to connect to a Keysight 34450A multi-meter for DC voltage measurement. The PCBs were suspended 5 cm away from a 2.4 GHz antenna connected to a Rigol DSG836 signal generator, as shown in Figure 10 and Figure 11. This setup was created to minimise any coupling between the transmitting antenna and the test wires; for example, the test wires were connected below the PCB as it was found that having the test leads close to the transmitting antenna increased the recorded DC result.

The measurement process involved incrementing the frequency from 1.8 to 3 GHz at 20 dBm transmission power and manually recording the DC voltage at each point. The results of the two measured rectennas are shown in Figure 12a. The 20 dBm transmission power represents the maximum output power available on the signal generator. Considering a path loss of 20 dBm [13], the received power of the rectenna would be at the target of 0 dBm, at which the rectenna was optimised to operate. Both plots in Figure 12a show peak voltages at 2.35 GHz of 0.97 and 0.84 V, respectively. This slight shift in resonant frequency was expected based on the prior S-parameter measurements of the antennas, but good results of 0.79 and 0.68 V were still recorded at 2.4 GHz.

As it was found that the resonance of these rectennas is at 2.35 GHz, a power sweep at varying distances was completed, as shown in Figure 12b. The power was then recorded at intervals of 2 dBm from 20 to 0 dBm and repeated at 7 and 10 cm.

The plot shown in Figure 12b shows an exponential relationship between power (in dBm) and the measured output voltage. This is the expected result and shows that the rectifier and the rectenna work well as a pair.

Based on the resistor measurements of 96 Ω/sq, the integrated resistor can be calculated to have a total resistance of 48 kΩ. Using Ohm’s law, the peak voltage of 0.97 V equates to 19.6 μW or −17 dBm of received power. Considering the whole system, this equates to an RF to DC efficiency ($P_{\mathrm{DC}}/P_{t}$) of 0.02%, or −37.1 dB. Using the data from the power sweep, the efficiency can be plotted from 0 to 20 dBm for 5, 7, and 10 cm in Figure 13. This shows that the efficiency of the rectenna begins to saturate at 20 dBm, but a higher efficiency could be expected if higher transmission powers were used.

To compare these results to similar rectennas reported in the literature, Table 2 was constructed from the rectennas identified in the introduction; all rectennas listed feature an on-chip antenna and operate at RF frequencies. The table lists some key performance data identified for the device, and the calculated figure of merit proposed in [18] is given by
(4)$$\mathrm{FOM}=\frac{\eta \left(\%\right)\times S^{3}}{A^{3/2}}$$
where η is the efficiency of the rectenna as a percentage, *S* is the separation distance between the transmitting and receiving antenna in mm, and *A* is the area of the rectenna in mm^2^.

The efficiency of the rectenna is the ratio of the power transmitted from the signal generator to the output power from the rectenna. This ratio is presented in dB as the power transfer efficiency (PTE) in Table 2. Alternatively, it could be converted to a linear value to be presented as a percentage, as this is what is used in (Equation 4).

However, there are some known limitations on this figure of merit. The first is that it is only valid for mm size rectennas, and the second is that it is designed for separation distances in the order of 10 mm. As such, this figure of merit would not be valid for the works from [17,22]. However, comparing rectennas becomes challenging due to differences in technologies, frequencies, and target applications. Therefore, this figure of merit serves as a reliable method to validate the performance of a rectenna. In this regard, it demonstrates that the designed rectenna performs well when compared to the existing literature. Additionally, by optimising the load to maximise the DC power, even higher power transfer efficiency could be achieved.

We can now compare the measured results of this work, which demonstrate a power transfer efficiency of 0.02%, or −37.1 dB. This is considered good given the die size, frequency, and test conditions. The integrated load of the rectenna could be optimised to increase the DC power, should a higher power level be required; however, a power delivery of 19.6 μW should prove sufficient for batteryless medical implants, particularly when combined with ultra-low-power sensors such as a 10.5 μW neural sensor [43] or a 5.5 μW capacitive sensor [44].

## 4. Discussion

This work reported the design, simulation, and measurements of a fully integrated miniature 2.4 GHz rectenna. The main component of the design was the antenna which aimed to have a large complex impedance, such that a matching network was not required. The design used a spiral with an integrated resistive matching stub to realise this aim. The design was optimised to minimise the reflection coefficient between the antenna and the rectifier circuit, with a simulated $S_{11}$ of −23.5 dB when the power received is 10 dBm.

The rectifier circuit made use of a novel tunnelling diode known as the ASPAT diode, which has strong temperature independence, zero-bias operation, and a high dynamic range. This circuit layout was designed to be a general voltage double such that the full wave of the input signal is rectified.

The antenna design was initially prototyped in CST, and the full rectenna was simulated using ADS momentum. The momentum simulations allowed for the optimisation of the design to determine an optimum antenna geometry and die size, with the aim to achieve a good impedance match at 0 dBm. The optimisation concluded with a reflection coefficient of −11.7 dB at 0 dBm, with an improved reflection coefficient of −23.5 dB at 10 dBm. The optimised geometry was simulated to have a far-field gain of −24.7 dBi, which is typical for an electrically small antenna at this scale.

The designed rectenna was then fabricated using i-line photolithography and wet etching processes, and then subsequently measured using DC and S-parameter measurements. The S-parameter measurements showed that the antennas did resonate, although at a lower magnitude than simulated. This still provided good DC measurements; 0.97 and 0.84 V peak voltages were obtained for two of the measured rectennas. As the resulting load resistor is 48 kΩ, these recorded values correspond to a power delivery of 19.6 and 14.7 μW for the respective device.

To conclude, a 4 × 1.2 ${\mathrm{mm}}^{2}$ rectenna was designed for the 2.4 GHz industrial, scientific, and medical band, which was sized appropriately for a 15-gauge needle. This rectenna was optimised to have a high impedance, thus eliminating the need for a matching network with a simulated reflection coefficient of −11.7 dB at 0 dBm received power. Through measurements, it was shown that this design can deliver a peak power of 19.6 μW at 0.97 V when positioned 5 cm away from a 2.35 GHz signal generator set to a power of 20 dBm.

## Figures and Tables

**Figure 1 sensors-23-06409-f001:**
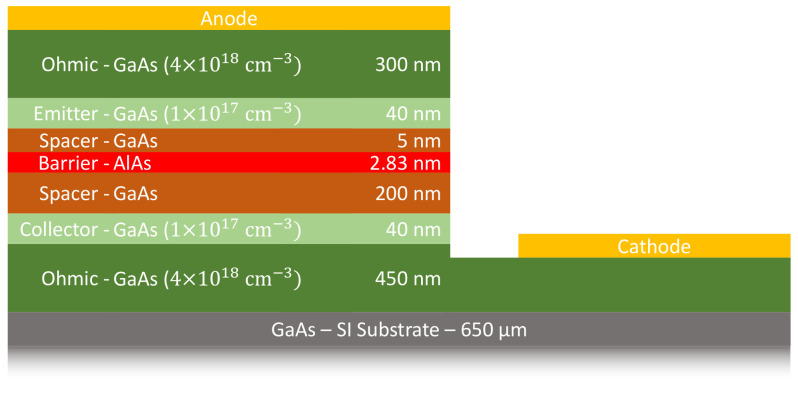
Cross-section of the ASPAT diode.

**Figure 2 sensors-23-06409-f002:**
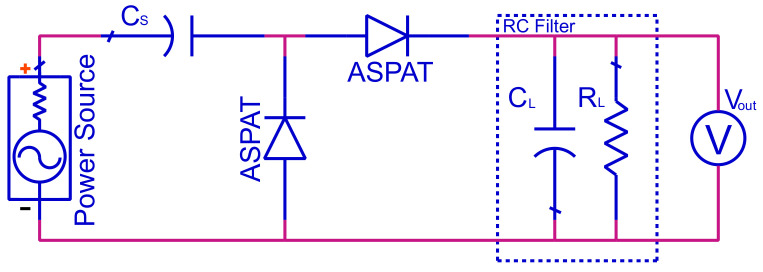
Schematic of a voltage doubler using two ASPAT diodes.

**Figure 3 sensors-23-06409-f003:**
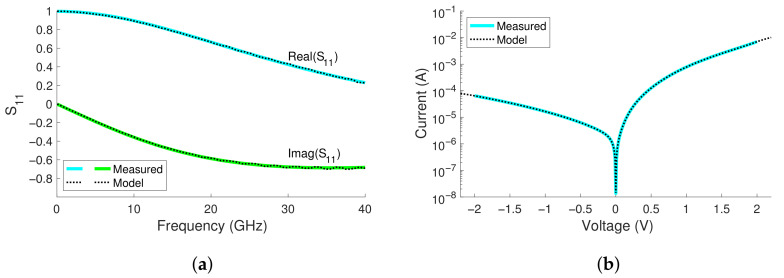
A comparison between the measurements and equivalent circuit model of the ASPAT diode using (**a**) S-parameters and (**b**) a DCIV sweep.

**Figure 4 sensors-23-06409-f004:**
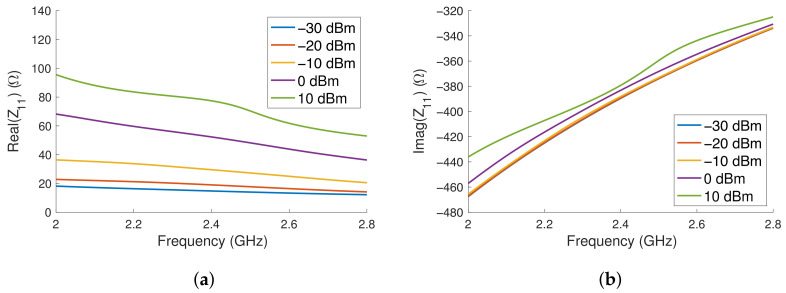
The simulated (**a**) real and (**b**) imaginary impedance of the rectifier.

**Figure 5 sensors-23-06409-f005:**
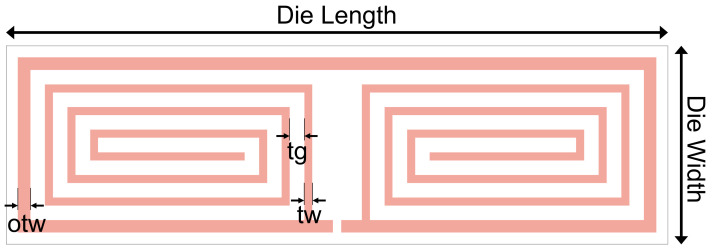
The trace width (tw), outer trace width (otw), trace gap (tg), die width, and die length labelled on a 2D layout.

**Figure 6 sensors-23-06409-f006:**
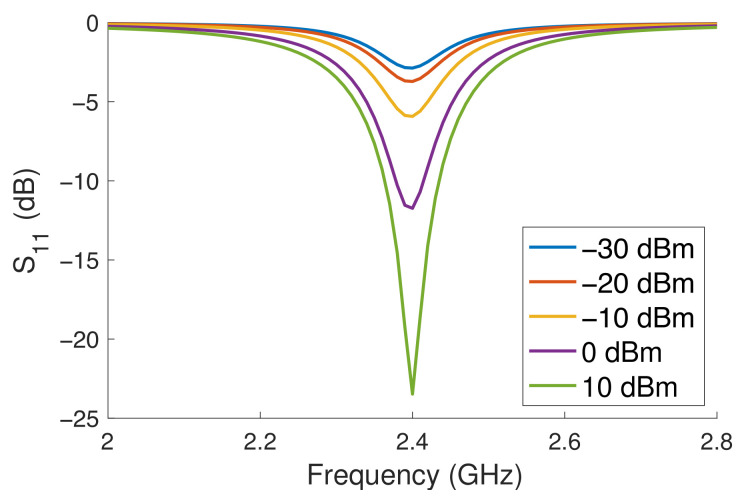
Simulated S-parameter results of the antenna and rectifier.

**Figure 7 sensors-23-06409-f007:**
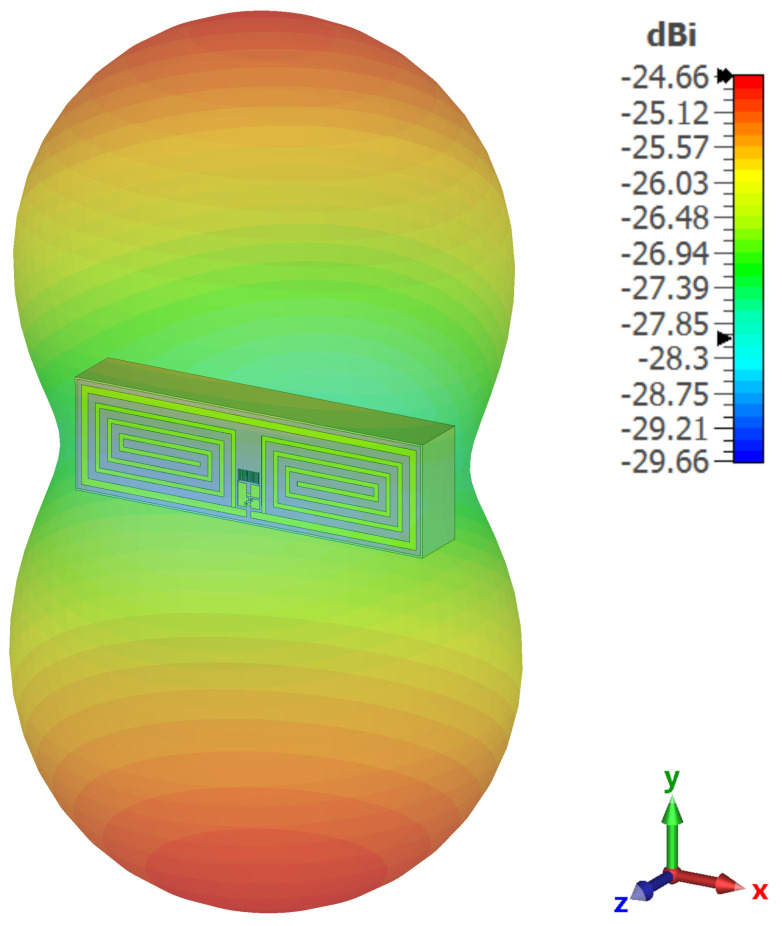
Far-field simulation results of the antenna in CST Studio.

**Figure 8 sensors-23-06409-f008:**
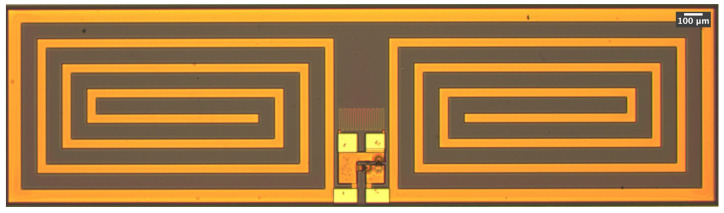
Die micrograph of the fabricated rectenna.

**Figure 9 sensors-23-06409-f009:**
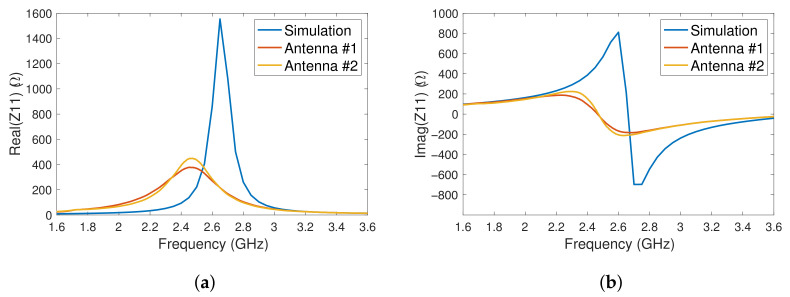
The simulated and measured (**a**) real and (**b**) imaginary impedance of the antenna.

**Figure 10 sensors-23-06409-f010:**
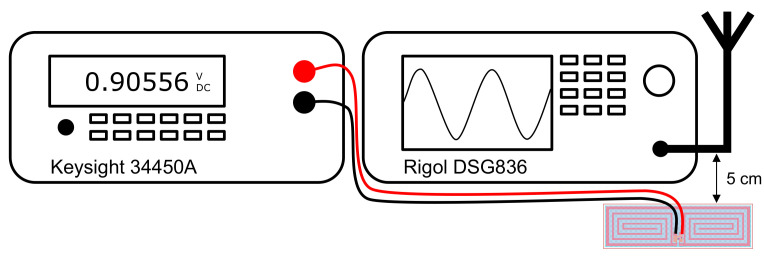
Block diagram of the measurement setup.

**Figure 11 sensors-23-06409-f011:**
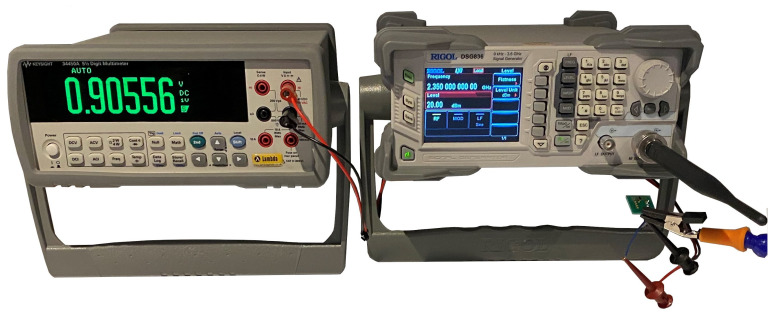
Photograph of the measurement setup.

**Figure 12 sensors-23-06409-f012:**
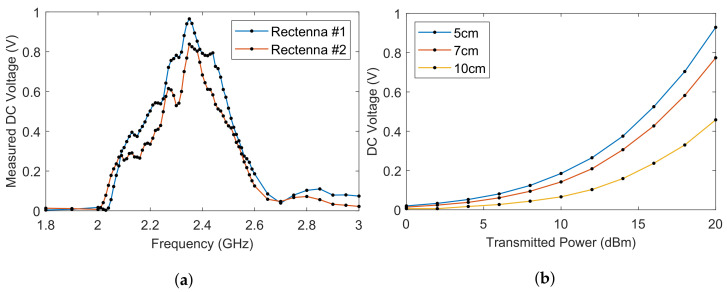
The measured results of (**a**) a frequency sweep from 1.8 to 3 GHz at 5cm and (**b**) a power sweep of 0 to 20 dBm at 5, 7, and 10 cm.

**Figure 13 sensors-23-06409-f013:**
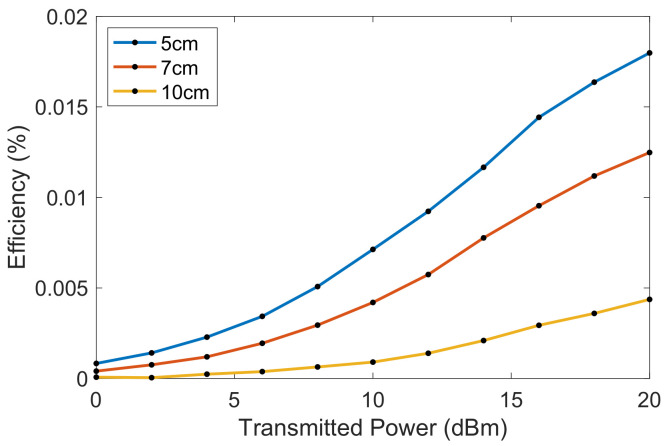
The derived power transfer efficiency at 5, 7, and 10 cm at 2.35 GHz.

**Table 1 sensors-23-06409-t001:** Comparison of diode parameters at 10 GHz [26].

Diode	Tss	$\beta_{v}$	$P_{\mathrm{sat}}$	Dynamic Range	ΔV
	(dBm)	$\left({\mathrm{VW}}^{-1}\right)$	(dBm)	(dB)	(%)
ASPAT	−55	5000	12	67	30
Si Schottky	−54	6000	8	62	120
PDB	−56	6000	10	66	70
Backward diode	−52	3000	−11	41	10

**Table 2 sensors-23-06409-t002:** Comparison with rectennas in the literature using on-chip antennas.

Ref.	Freq. (GHz)	Antenna	Rectifier	Area (mm^2^)	S ^a^	Voltage (V)	P*_t_* ^b^ (W)	P_*DC*_ ^c^ (μW)	PTE ^d^ (dB)	FOM ^e^
[14] (2008)	2.45	Coil	CMOS	1 × 0.5	0.05	0.6	0.25	94.7 ^f^	−34.2 ^f^	0.01^f^
[15] (2010)	5.8	Folded with Stubs	CMOS	3 × 1.5	7.5	1.8	4	0.6 ^f^	−68.3 ^f^	0.66^f^
[16] (2013)	5.2	Monopole	CMOS	3.2 × 1.5	3.5	1.15	5	100	−47.0 ^f^	8.13 ^f^
[17] (2014)	24	Dipole	CMOS	3.7 × 1.2	28	0.9	10	1.5	−68.2 ^f^	35.2 ^f^
[18] (2014)	0.16	Coil	CMOS	2 × 2.18	1	N/A	N/A	N/A	−18.5 ^f^	156 ^f^
[19] (2016)	1.1	Folded Dipole	CMOS	0.7 × 0.62	2	1	1	50	−43.0 ^f^	142 ^f^
[20] (2017)	2.75	Loop	CMOS	1.6 × 1.6	2	1.1	1	1200	−29.5	221 ^f^
[21] (2022)	0.4	Coil	CMOS	2 × 2	1.6	N/A	0.4	53.2	−38.8 ^f^	6.81 ^f^
[22] (2022)	28	Patch	pHEMT	1.8 × 1.8	500	0.19	25,000	8.6	−94.7 ^f^	734 ^f^
[13] (2022)	2.57	Folded Spiral	ASPAT	1 × 5	5	0.8	0.1	6.4	−41.9 ^f^	71.6 ^f^
This Work	2.35	Folded with Stubs	ASPAT	1.2 × 4	5	0.97	0.1	19.6	−37.1	233

^a^ Separation distance between transmitting and receiving antennas. ^b^ Transmitting antenna power. ^c^ Calculated DC output power. ^d^ Power transfer efficiency. ^e^ Figure of merit [18]. ^f^ Result was calculated from the available information.

## Data Availability

Data sharing not applicable. No new data were created or analyzed in this study. Data sharing is not applicable to this article.

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
