# Peer review of "Miniature Integrated 2.4 GHz Rectennas Using Novel Tunnel Diodes"

_sensors, 2023, doi:10.3390/s23146409_

Round 1
Reviewer 1 Report
In this paper the authors present the design, fabrication, and characterization of a fully integrated miniature rectenna using a novel tunnel diode known as the Asymmetrical Spacer Layer Tunnel. The antenna is fully integrated with the rectifier on a single chip, thus enabling antenna miniaturization due to the high dielectric constant of GaAs and spiral design. This miniaturization enables the design to be fabricated economically on a GaAs substrate , thus unlocking applications in medical implants. The design presented here has a total die size of 4 x 1.2 mm2, with a maximum measured output voltage of 0.97 V with a 20 dBm single tone 2.35 GHz signal transmitted 5 cm away from the rectenna.
The development of new designs for rectennas is a very interesting topic and the miniaturization proposed here sounds nice. Anyway, the manuscript in the current state is really confused and hard to be read and understood. In order to be published it needs a radical improvement. Below my comments:
1. it is important to define all the acronyms used in the text (see line 55, RFID; line 59 pHEMT; line 75 MMIC)
2. line 143 (tatanium nitride (Ti??))
3. line 146-149. this paragraph is really hard to be understood. please write it in a more clear way
4. sectoin 2.4- fabrication. it is not clear, reading the text, how the fabrication proceed. A figure could help a lot. (for example: "of which there are 2 etch steps, 5 dielectric layers, and 4 metal layers", to me means almost nothing....)
5. figure 7 - please add a scale bar
6. figure 12- please correct the caption
7. line 304 . "outputted"?
One of the main point in this paper, according to my understanding, is the use of Asymmetrical Spacer Layer Tunnel diode. anyway there is ZERO details on this diode design, structure and fabrication. On the contrary the authors report details on an obvious spiral antenna. This must be improved. A figure describing the ASLT and how it is used here can help the reader and could be critical for the publication.
Finally, I think that it could be useful to better discuss the state-of-the-art in rectenna technology. There are several recent important papers on the topic that can be mentioned and could help the reader to understand the importance of this research field, for example:
-Science 2020, 367, 1341.
-ACS Appl, Nano Mater. 2021, 4, 2470.
-Nat. Nanotechnol. 2015, 10, 1027.
-Adv. Energy Mater. 2022, 2103785
-Adv. Electron. Mater. 2018, 4, 1700446.
-ACS Appl. Electron. Mater. 2019, 1, 692.
the english language needs a revision. there are several errors along the text, but I think that this is not a major issue
Author Response
Dear Reviewer,
We would like to express our gratitude for reviewing our academic paper. We appreciate your valuable feedback and suggestions for improvement. In response to your comments, we have made the necessary changes to enhance the clarity and comprehensibility of the manuscript. We address each of your points below:
1. Acronyms: We have now included definitions for all the acronyms used in the text (see lines 58-68), including RFID, pHEMT, and MMIC. Additionally, we have added MMIC to the list of abbreviations for better reference.
2. Line 161: We have corrected the mention of "titanium nitride (Ti)" to "titanium (Ti)".
3. Line 163-167: We have rephrased it to improve its clarity and readability.
4. Section 2.4 - Fabrication: We understand your concern about the lack of clarity regarding the fabrication process. Since this paper primarily focuses on the design and characterization aspects rather than fabrication, we have simplified the paragraph. We have also included a reference to a more detailed process of the Asymmetrical Spacer Layer Tunnel (ASPAT) for readers who seek additional information. Furthermore, we have defined the layers used for capacitors, resistors, and the antenna.
5. Figure 7: We appreciate your suggestion. We have added a scale bar to Figure 7 for better visual reference.
6. Figure 12: We apologize for the error in the caption. It has been corrected accordingly.
7. Line 319: We have revised the sentence as you suggested to improve clarity. It now reads, "The efficiency of the rectenna is the ratio of power transmitted from the signal generator to the output power from the rectenna."
Regarding your point about the Asymmetrical Spacer Layer Tunnel (ASPAT) diode. We agree that this is an essential aspect of our work. To address this, we have added a description of the ASPAT's structure with citations and included a cross-section figure for better visualization, please see lines 80-89.
Finally, we appreciate your suggestion to discuss the state-of-the-art in rectenna technology more thoroughly. We have now acknowledged examples of optical rectennas in the introduction and cited four of the suggested papers to help readers understand the importance of the research field, please see lines 26-32.
Once again, we sincerely thank you for your time, effort, and valuable feedback. We believe that the revisions we have made based on your suggestions have significantly improved the quality and clarity of our manuscript. We hope these changes meet your expectations and look forward to hearing your further comments.
Best regards,
Chris
Reviewer 2 Report
sensors-2443950 represents an ingenious technological development and of clear potential value for its applicability. Very well conceived, it is written with crystal clarity and very aptly illustrated. I only have two recommendations for authors: (1) offer a more concise version of the Key words. (2). After the discussion, incorporate as point 5. A brief paragraph of Conclusions, highlighting 'the lemon juice' that they have squeezed.
Author Response
Dear Reviewer,
We would like to express our gratitude for reviewing our academic paper. We appreciate your valuable feedback and suggestions for improvement. In response to your comments, we have made the necessary changes to enhance the clarity and comprehensibility of the manuscript. We have carefully considered your recommendations and have made the following revisions accordingly:
Concise version of the Key Words: To address this, we have removed some keywords that were similar to others used, ensuring a more streamlined and effective representation.
Conclusion: In response to your suggestion, we have incorporated a brief paragraph of conclusions as a separate point after the discussion section on line 364. This paragraph highlights the significance of our research and the key findings that have been obtained.
Once again, we sincerely appreciate your positive feedback and constructive recommendations. We believe that the changes we have made based on your suggestions have significantly improved the paper. We hope these revisions meet your expectations, and we eagerly await any further comments or suggestions you may have.
Thank you for your time and valuable input.
Best regards,
Chris
Reviewer 3 Report
1. The article is within the scope of the journal. The authors presented a well-organized paper and easy-to-read and to-follow.
2. In this paper, the authors have presented the design of a fully integrated miniature rectenna using a tunnel diode known as the asymmetrical spacer layer tunnel.
3. The authors have designed an antenna to be impedance matched with the rectifier, eliminating the need for a matching network and saving valuable real estate on the GaAs substrate.
4. To validate the theoretical results of the proposed rectenna, the authors have carried out measurements using two prototypes of the fabricated rectennas. Measured results have well supported the theoretical findings.
5. The authors have compared the proposed rectennas performance with those published in the literature to reveal the robustness of the proposed one.
6. There is a clear methodology. There is an extensive explanation of the method and a discussion of the results.
However, I suggest the authors address the following issues:
1. Return loss in dB is positive, and the reflection coefficient is negative. Please, take care of the terms and their physical meaning. The term (Return loss) should be replaced, wherever it is mentioned in the text, by either (input reflection coefficient) or only (S11). For more information, the authors are advised to see:
T. S. Bird, "Definition and Misuse of Return Loss [Report of the Transactions Editor-in-Chief]," in IEEE Antennas and Propagation Magazine, vol. 51, no. 2, pp. 166-167, April 2009.
doi: 10.1109/MAP.2009.5162049
2. The language of the article has to be slightly revised. The attached file contains many suggestions and comments on the first two pages.

The language of the article has to be slightly revised. The attached file contains many suggestions and comments on the first two pages.
Author Response
Dear Reviewer,
We would like to express our gratitude for reviewing our academic paper. We appreciate your valuable feedback and suggestions for improvement. In response to your comments, we have made the necessary changes to enhance the clarity and comprehensibility of the manuscript. We have carefully considered your recommendations and have made the following revisions accordingly:
We have replaced all instances of "return loss" with either "reflection coefficient" or "S11" wherever appropriate.
We appreciate your feedback on the language used in the article. We have reviewed the attached file containing your suggestions and comments, and we have incorporated the proposed changes into the revised paper. Additionally, we have made similar adjustments throughout the document. However, we have retained the use of British English, including terms such as "optimising" instead of "optimizing," for coherence with the rest of the text.
Once again, we sincerely appreciate your positive feedback and constructive recommendations. We believe that the changes we have made based on your suggestions have significantly improved the paper. We hope these revisions meet your expectations, and we eagerly await any further comments or suggestions you may have.
Thank you for your time and valuable input.
Best regards,
Chris
Round 2
Reviewer 1 Report
the authors considered all my comments and improved the manuscript according to them.
I recommend the publication
the manuscript needs some check, but it is almost ok